# Effect of Instability Training on Compensatory Muscle Activation during Perturbation Challenge in Young Adults

**DOI:** 10.3390/jfmk8030136

**Published:** 2023-09-15

**Authors:** Stephen C. Glass, Kamryn A. Wisneski

**Affiliations:** Human Performance Lab, Department of Movement Science, Grand Valley State University, Allendale, MI 49401, USA; wisneska@mail.gvsu.edu

**Keywords:** instability training, slosh tube training, electromyography, neuromuscular training, force steadiness

## Abstract

Balance requires constant adjustments in muscle activation to attain force steadiness. Creating appropriate training can be challenging. The purpose of this study was to examine the effects of 2 weeks of front squat instability training using a water-filled training tube on force steadiness during an instability challenge. Control (CON, *n* = 13) and experimental (EXP, *n* = 17) subjects completed pre- and post-testing for EMG variability by completing one set of 10 repetitions with a stable and unstable training tube. Electrodes were placed bilaterally on the anterior deltoid, paraspinal, and vastus lateralis muscles. CON subjects completed 2 weeks of training using a stable training tube, while EXP subjects trained with a water-filled instability tube. EMG data were integrated for each contraction, and force steadiness was computed using the natural log of coefficient of variation. CON results showed no changes in force steadiness for any condition. EXP showed significant reductions in EMG activation variability across all muscles. These results indicate a significant training effect in reducing muscle activation variability in subjects training with a water-filled instability training device. Improvements seen in these healthy subjects support the development of training implements for a more clinical population to help improve force steadiness.

## 1. Introduction

Balance and related falls may be due to a combination of neurologic conditions but also detraining from inactivity and aging. Sarcopenia, weakness in postural muscles due to a lifestyle of prolonged sitting, and low back pain result in reduced compensatory muscle activation [1,2,3,4]. These changes in activation timing and recruitment pattern can result in a loss of postural control and increased risk of falls or injury [5]. Balance improvements can be achieved by disrupting the neuromuscular system so that it is forced to adjust in postural stability and muscle activation—otherwise known as compensatory adjustments. Stable posture is achieved by making multiple minor adjustments as opposed to large rapid motions [2]. While falling in older adults is a commonly identified clinical issue, there can be a range of stability impairments in both clinical and nonclinical populations, and, therefore, balance training may be viewed in the context of habilitation to return to optimal function as well as rehabilitation due to a clinical condition.

A large body of works in the literature have examined the effects of instability training on the changes in muscle activation and strength [6,7,8,9,10,11,12]. Typically, the percent of maximal voluntary contraction was assessed with varied types of instability challenges. These challenges have been grouped according to the location of the instability. Unstable surfaces, such as a Swiss ball [13,14,15,16,17,18], Bosu ball [7], TRX bands, wobble boards, and surfaces that may vary, are termed “bottom up” instability devices, while loads that are carried and affect the stability of the upper body are termed “top down” devices. Other devices are specific to a single limb, such as Bodyblade trainers [19,20]. A common theme among the studies suggests that when the instability is bottom up, there is a redistribution of muscle activation to the core and supporting muscles. Marshall and Murphy [17] had subjects perform bench press exercise at 60% 1RM on a stable bench and a Swiss ball bench. They found increased deltoid and abdominal activity on the less stable Swiss ball, suggesting greater core and limb stability activation. Other studies have shown similar results using plank exercise [21], balance boards [22], and Bosu or Swiss ball exercise [12,13,14]. However, just activating core musculature may not result in improvements in postural control. In fact, research has suggested that lifting stable loads allows one to lift heavier loads and actually activates more core musculature than with unstable loads [6,8,9,11,18]. Anderson and Behm [11] showed reductions in peak force when subjects lifted on an unstable surface, and Hamlyn et al. [10] showed that core muscle activation was significantly greater with 80% 1RM squat and deadlift compared to body weight instability exercises.

Stability during movement is not solely related to the amount of strength that can be deployed by a given muscle. Balance loss and even a fall results from an inability to recover from a trip, sudden motion challenge (slip), or impact (bump or push). Managing the balance challenge starts with proprioceptive inputs sensing body position and efferent outputs to agonist and antagonistic muscles to allow for appropriate balance correction. In older adults, falls may be due to limitations in neuromuscular performance. Claudino et al. [2] examined the timing of postural adjustments following body perturbations induced by way of a ball impacting laterally at shoulder level in older adults, with and without a fall history, as well as young adults. In the fall subjects, they noted a longer latency in activating stabilizing muscles as well as a longer time to peak displacement of the center of pressure. Thelan et al. [1] induced a sudden forward lean necessitating a forward step to regain balance in young and old adults They identified an age-related slowing of activation and activation of agonist–antagonistic muscles, resulting is stiffer joint movement, which could contribute to fall risk. Additionally, pain associated with a chronic condition, such as low back pain, may also contribute to disruptions in the response to some type of balance challenge. Hedayati et al. [3] examined subjects with a history of low back pain. At the time of testing, they were pain-free and not fatigued; yet, when given a postural challenge, they exhibited an impairment in the expected postural adjustments. Specificity of training would dictate that to train for stability, there must be perturbations that train the neuromuscular system to adapt to both proprioception inputs to a destabilizing force as well as the neuromuscular coordination of the efferent muscular response. While there are several methods of creating unstable surfaces (bottom up) for instability, there are limited methods and investigations of upper body perturbations that would force postural muscle compensation. Nairn et al. has shown that the location of the instability will influence the location of compensatory muscle activation [23,24], and studies are also suggesting that another way to examine stability is examining the variability of activation, sometimes called “force steadiness”. Force steadiness can be obtained by reducing the latency between destabilization and muscle reactivity [25], improving joint proprioception [26]. Perturbation balance training consists of destabilizing forces that force repeated muscular contractions to maintain stability [27]. Perturbations may be manually applied, as in a sudden bump, or lifting a load that induces destabilizing forces [28,29,30,31,32]. The limitations to these methods are that they are one-time, abrupt challenges that may provide a reasonable test situation but may not be reasonable for training. While there is substantial research regarding the effect of perturbation balance training using the bottom-up training method [33,34,35,36,37,38], there are limited studies examining upper body perturbation training programs on improvements in force steadiness that provide both a continuous proprioceptive challenge requiring continual compensatory muscle activation.

One training tool used to induce upper body perturbation is a water-filled training tube known as a “slosh tube”. Most training tubes are simple cylinders partially filled with water and are available for commercial purchase. Often these are used in health and fitness settings for nonclinical exercisers. During a lift, the water creates inertial movements in a variety of directions, forcing support muscles to rapidly compensate. Novel water-filled training tube studies by Glass et al. [31,32] used a tube that was fitted with a central valve that could be manipulated, creating different flow variations. In one study [31], subjects completed bicep curls with different valve settings (open flow, 45-degree angle obstruction to flow, and no flow control) and found significant variability in muscle activation in the paraspinal and deltoid muscles during both concentric and eccentric contractions. Rather than examining the amount of muscle activated, the exercise was performed with an 11.4 kg tube and the log of the coefficient of variation was determined as a marker of force steadiness. A similar study was performed using an overhead squat [32] and showed that as the valve setting was changed, the muscles that showed the most reactivity and instability also changed. These studies showed that water-filled tubes create perturbations due to turbulence and can induce the variability of muscle contraction that may serve as a neuromuscular training tool.

Water-filled tubes can induce subtle changes in load forces, such that proprioception is challenged, necessitating compensatory muscle activation. To date, little research is available on whether training with these tubes can reduce the variability in muscle contractions using instability training. The purpose of this study was to examine the effects of 2 weeks of instability training using a water-filled training tube on force steadiness during an instability challenge.

## 2. Materials and Methods

Subjects were recruited from a university population. They were healthy, active, and free of any skeletomuscular condition or injury that might impair their ability to complete the study. Subjects exercised 2–3 days per week, and most were aerobic and recreational sport-conditioned. It was not required that they have weightlifting experience, since the lifting load was light, and a familiarization period was provided. The study was conducted according to the guidelines of the Declaration of Helsinki and approved by the Institutional Review Board of Grand Valley State University (protocol code 20-015, approved 29 July 2019). Based on most recent published studies using the same device [31,32] with a power of 0.80 and effect size of 0.40, given a control and experimental group, a target size of 30 (15 control, 15 experimental) subjects was projected to have adequate power for analysis. This study took place across January 2019 and May 2023, which was the time of the COVID pandemic. As such, we had some subjects lost due to stoppage of research activities. A total of 2 control subjects withdrew, and our randomization dictated that the last subjects selected were experimental, giving us a total of 13 control subjects, 17 experimental subjects, and 30 total subjects. We were not approved by IRB to exceed 30 subjects, so we concluded with an unbalanced sample size. Subjects’ characteristics are shown in Table 1.

### 2.1. Day 1 Assessment

After providing signed informed consent, subjects completed a health history questionnaire as part of initial assessment. Height, body mass, and resting seated blood pressure were recorded. Subjects with resting blood pressure over 130/90 were retested after a brief rest and excluded for participation if values remained above the cutoff criteria. Following initial screenings, subjects were randomly assigned to either the Control (CON) or Experimental (EXP) group. Randomization was completed using a random number generator https://www.graphpad.com/quickcalcs/randomize2/ (accessed on 11 September 2019). Subjects were then provided a familiarization period with the water-filled instability tube as well as the front squat movement. Two spotters were used to place the tube into the hands of the subjects in the front squat position (Figure 1) and a metronome timer was set to pace a 17 repetition-per-minute pace. Subjects practiced with 5 to 10 repetitions with a stable tube as well as the unstable tube.

Following the familiarization period, subjects were prepared for electromyographic (EMG) electrode placement. Subjects’ skin was shaved and cleaned with alcohol, after which, 6 pre-amplified surface electrodes (Biopac Systems, TSD150B, 20 mm interelectrode distance) were placed bilaterally over the muscles of the anterior deltoid, paraspinal, and vastus lateralis muscles in locations established by Cram [33]. Flexible tape was used to secure electrodes to the skin and still allow freedom of movement. Following subject preparation, all subjects completed the initial testing for EMG steadiness by completing one set of 10 repetitions with the stable tube followed by the unstable tube. EMG data were recorded at 2000 samples per second using a Biopac MP160 system (Goleta, CA, USA) and Acknowledge software (Biopac version 5.0). Subjects were paced on each contraction using a metronome set at a 17 rep/min cadence. Due to the unstable nature of the squat when partially filled with water, a manual (visual) marking system was used to determine the transition between concentric and eccentric contractions.

### 2.2. Training Tube Specifications

The instability tube was constructed of high-density plastic material, with screw caps on each end to allow for water addition. The tube dimensions (length: 159.4 cm, diameter: 11.4 cm, circumference: 36.2 cm) necessitated straps be attached as support for securing the tube during the squat maneuver (Figure 1). Dry weight of the tube was 5.0 kg, and, for the study, 6.0 L of water was added for a maximum weight of 16.0 kg. A volume was chosen that maximized both water movement and adequate load stimulus. The tube was also fitted with an adjustable central valve, which could be set at a 45-degree position to provide water turbulence in line with the spinal column during the squat (unstable EXP setting). The valve could also be closed to prevent movement of water across the tube (stable CON setting).

### 2.3. Training Days

Both CON and EXP subjects completed 6 exercise sessions with their respective tube. Training days were set a minimum of 48 h apart. If a day was missed, it was rescheduled within a week to ensure all subjects completed 6 sessions of training. Subjects reported to the training lab having not done any other exercise that day. Subjects warmed up on a cycle ergometer at 25–30 watts for 5 min. Following the cycling, subjects completed 2 sets of 10 body weight squats with 3 min of rest between each set. Following this warmup, subjects completed 4 sets of 15 repetitions with their respective tube condition (CON Stable, EXP unstable).

### 2.4. Post-Testing

On the last test day, subjects repeated the initial 10-repetition test for both the stable and unstable tube, with EMG data collected bilaterally on the anterior deltoid, paraspinal, and vastus lateralis muscles. There was no familiarization trial on the post-test day.

### 2.5. Data Processing

All raw EMG data were filtered using a filter suggested by the manufacturer (Biopac systems) to remove ECG waveforms observed in the core (paraspinal) muscles. This was a high-pass filter, Blackman −67 db, with a 30 Hz cutoff frequency and 255 coefficients. The pre-amplified electrodes also had a 58–61 Hx band-stop filter to remove background noise from electrical lighting. Data were then rectified, and each concentric and eccentric contraction were individually integrated (IEMG) for all repetitions and all muscles. The means and SD of the IEMG for concentric and eccentric contractions were computed across all muscles for pre- and post-training trials, for stable and unstable squat settings. Force steadiness is essentially the variation in the integrated EMG measures across repetitions. Past research has used standard deviation as a measure [3], coefficient of variation [37], and log of the CV [31,32]. We chose the latter to match the methodology of our previous work. Using the integrated EMG signal across test trial repetitions, force steadiness was measured using the natural log of the percent coefficient of variation (Ln (SD/mean)). To complete an analysis of variance (ANOVA) using CV, 3 assumptions needed to be met: the CV values were required be approximately normally distributed, the variances of the population were required to be equal, and the observations were required to be independent. Two assumptions were not me: the equal variance assumption and the normality assumption. This necessitated the use of the natural logCV.

### 2.6. Statistical Analysis

A 3-way analysis of variance was used to detect differences by group (CON vs. EXP), contraction type (concentric vs. eccentric), and condition (pre vs. post). This was performed for each muscle. Post hoc Tukey’s tests were used for paired comparisons in the presence of main effects.

## 3. Results

ANOVA results showed no main effect differences between concentric and eccentric contractions, so the results for contraction type were collapsed and data combined for analysis. As expected, the control group showed no post-training changes in activation variability for any muscles, across both stable and unstable tube tests. For the experimental group, significant post-training reductions in activation variability during unstable tube testing were seen in five of the six muscles studied. The experimental group showed no pre/post changes in activation variability for the stable tube test. The magnitude of reduction ranged between 9 and 17% in activation variability.

Figure 2, Figure 3, Figure 4, Figure 5, Figure 6 and Figure 7 show the activation variability results comparisons between the control and experimental groups. A significant training effect was seen for the experimental group for all muscles except the left vastus lateralis muscle. Instability training resulted in significant reductions in activation variability when presented with the unstable tube test. The control group did not show any changes in EMG activation variability pre/post training.

## 4. Discussion

The results of this study show significant reductions in the variability of muscle activation (improved force steadiness) during a destabilizing weight training challenge following 2 weeks of instability training. Healthy active subjects trained across six sessions with a water-filled instability tube showed significant improvements in force steadiness measured by EMG in the deltoids, paraspinal, and vastus lateralis muscles. Control subjects training with a stable tube showed no improvement in force steadiness. Balance and postural stability are not simply the result of muscle strength but, rather, proprioceptive inputs to sense the unstable forces, and the neuromuscular system’s ability to initiate small, rapid contractions of postural support muscles to maintain stability. Research is plentiful with studies examining the effect of instability training on improvements in muscle activation [6,7,9,10,12,13,14,15,17,20,21,23,24,29,30]; however, balance and postural stability require more extensive coordination of proprioception and the motor cortex. Few have examined improvements in the compensatory activation of muscles to maintain force steadiness. Aviles et al. [34] studied older adults across a 4-week training program where subjects were exposed to trip-like perturbations on a treadmill. Qualitative interviews showed that participants perceived positive benefits from the training, but access and safety systems needed for such training were deemed as a possible limitation to using fall-inducing activities as a training program. A review by Behm and Colado [35] identified a number of training benefits from instability training but also identified that loads used for training need to be reduced due to the unstable support surface for much of instability training. The present study utilized a unique type of instability delivery that allows for a stable support surface without any sudden change in gait or balance, which may provide a safer means of instability training.

This study demonstrated that even in healthy active individuals, force steadiness can be improved with only 2 weeks of training. Training for improvements in the neuromuscular system may come in many forms. For muscle tissue growth, overload in the form of weight and fatigue of the myofibrils is essential for tissue hypertrophy. However, stability is also dependent upon the initial proprioceptive awareness of the instability, degree of compensatory adjustment [29], onset of firing (latency) [25], and coordination of firing [25,26,36,37] of various postural muscles. Often, these force adjustments are small yet rapid. Standard forms of strength training will not provide the stimulus for adaptation in this case. Instead, training specificity dictates that the load challenge be random, unpredictable, and involve a wide range of muscles for balance control. In fact, studies suggest that as instability increases, the training load that can be used is reduced [11,13]. Therefore, training loads needed to produce minor balance perturbations should be small, rapid, and random. One other interesting result of the present study is the lack of a significant improvement force steadiness in the left vastus lateralis. Unfortunately, we did not assess limb dominance in this study, but one might speculate that limb dominance could influence the degree of adaptation. Much of the unstable load is absorbed by the deltoid and paraspinal muscles, so it may be that the leg did not receive the same degree of training challenge. A study by Promsri et al. [38] found that postural acceleration challenges given while the participants balanced on their dominant leg showed higher postural control efficiency compared to nondominant. Limb dominance and stability adaptation may be an interesting topic for future work.

Imbalance can be initiated in the upper body as result of postural deficiencies, mass redistribution due to obesity, and physical perturbation by means of a bump or sudden change in direction. The present study examined muscles challenged by an upper body perturbation requiring a “top down” adjustment in muscle activation to maintain stability. Utilizing the water-filled tube, random and unstable loads were given during training, resulting in a degree of muscle activation variation. Significant adaptations to these perturbations were seen in deltoid, paraspinal, and vastus lateralis muscles, indicating a whole-body adaptation and improvement in force steadiness. Upper body instability has previously been induced using water-filled tubes. In a series of studies, Glass et al. [31,32] found that water-filled tubes with a central valve redirecting water flow in different planes created significant perturbation to upper body support musculature. Ditroilo et al. [29] used the squat exercise to induce instability and increased postural sway using a water-filled device along with associated activation of core musculature. As per the concept of specificity of training, adaptations to instability challenges can only be trained by giving the subjects unexpected perturbation. The training method used in the present study provides a more novel approach compared to traditional impact or trip-style perturbations.

Perturbation training is a relatively new concept, and studies have been examining different means of providing instability challenges. However, interest is increasing among clinicians [27,39,40,41]. Safely providing random instability challenges forcew patients to use proprioceptive, tactile, and visual cues to regulate balance. If the patients themselves are initiating movement that results in the random instability, as is the case with water-filled tube training, then the device can serve as effective biofeedback to help coordinate movement. Training with a water-filled tube means that the movement of the exercising individual is partially responsible for the degree of instability created. Unstable movement would exacerbate the forces of water and resulting instability. With the biofeedback linking movement to instability, the user begins to adopt more stable movement practices to reduce the amount of instability. Coupling the smoother movement with improved compensatory activation of posture support muscles thus results in improved force steadiness. Biofeedback is often visually linked, where the exercise monitors their movement by following a visual tracking of changes in the center of pressure [42]. This can sometimes be disorienting and is not utilizing any internal feedback sensations. Water tube training provides this immediate proprioceptive feedback, which may translate into a more manageable feedback system. In the present study, we utilized young, healthy individuals with no balance problem, yet still saw significant improvements in force steadiness. This is likely due to the effective biofeedback (proprioception) training that resulted in subjects executing smoother movements that resulted in less water movement and, therefore, less instability.

There are limitations to this study. The study length was limited to only 2 weeks of training, so any adjustments in stability are likely related to the neuromuscular system adjustments. It is not known how long the adaptation will last, nor whether the adaptation is transferable to other balance challenges. Our sample was young adults, so transferring the adaptation seen to an older population would be premature. However, future studies with older populations and even moderate-aged, detrained populations could utilize similar devices that are modified to be lighter and provide less vigorous load challenges. The advantages of using a water-filled pipe are that the length of the tube can be modified to increase or decrease the shifts in load past a given base of support, and the amount of water and degree of movement can be altered. Functional training involves a mix of training techniques to cause a range of neuromuscular adaptations. The advantage of a water-filled training tube is that the load is light and the perturbations relatively minor, avoiding unstable surfaces that can increase risk of falls during training. This allows more selective training of postural muscles during a function movement, such as walking, squatting, or other movements that involve posture stabilization. Another limitation of the present study is that the instability challenge was the same as the training, so the observed training effect may be very specific to the method training. Future studies should consider using this fluid tube-style training with older populations and compare a wider array of balance and functional measures to identify any functional training adaptations that may occur. However, it is promising to see that even in a young, healthy adult population, a brief exposure to this form of instability training results in significant improvements in force steadiness. This form of training may provide an additional means of training populations for improved stability.

## Figures and Tables

**Figure 1 jfmk-08-00136-f001:**
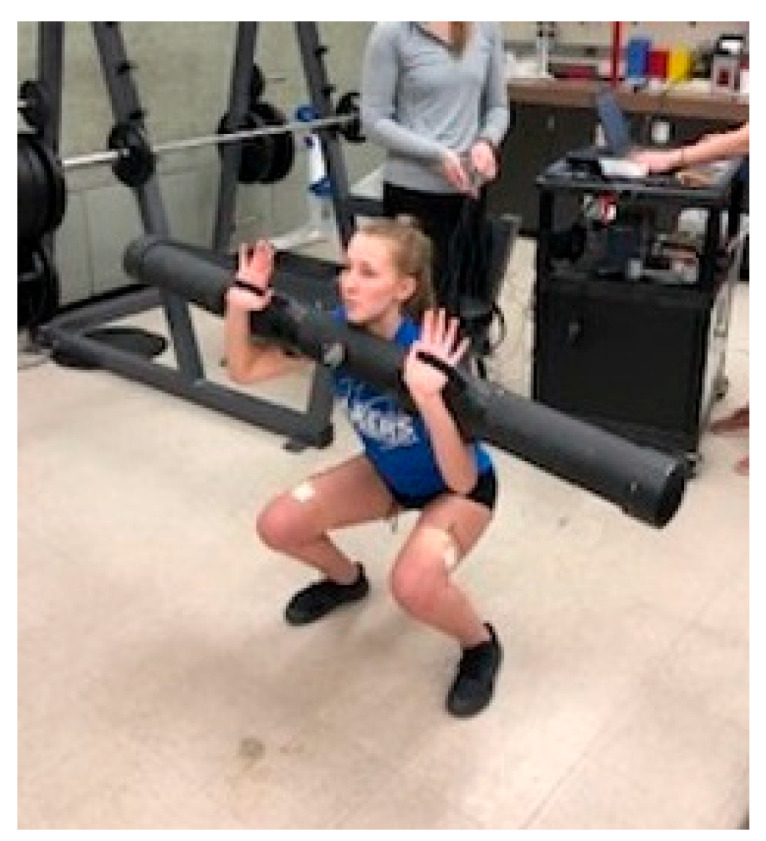
Subject front squat position with instability training tube.

**Figure 2 jfmk-08-00136-f002:**
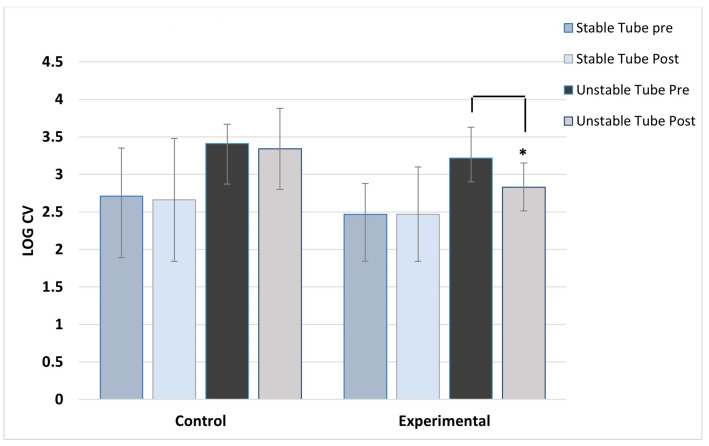
Right deltoid activation variability. ***** indicates the experimental group’s significant post-training reduction in activation variability with unstable tube.

**Figure 3 jfmk-08-00136-f003:**
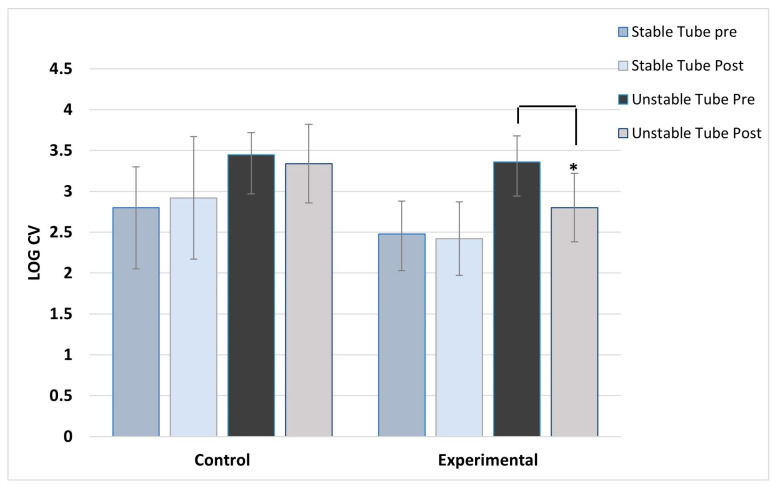
Left deltoid activation variability. ***** indicates the experimental group’s significant post-training reduction in activation variability with unstable tube.

**Figure 4 jfmk-08-00136-f004:**
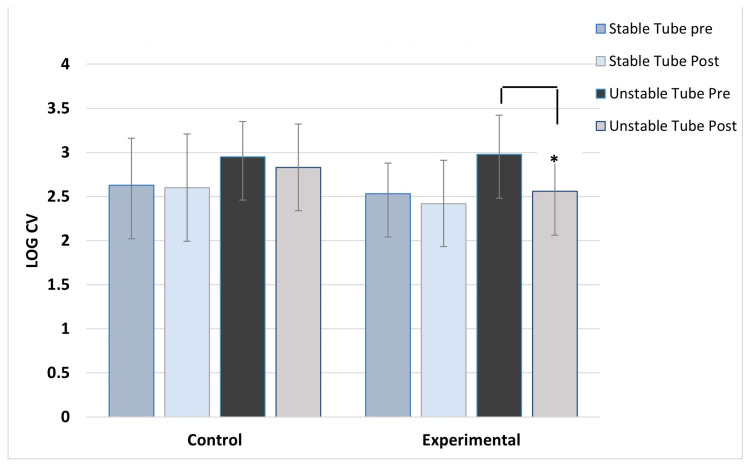
Right paraspinal activation variability. ***** indicates the experimental group’s significant post-training reduction in activation variability with unstable tube.

**Figure 5 jfmk-08-00136-f005:**
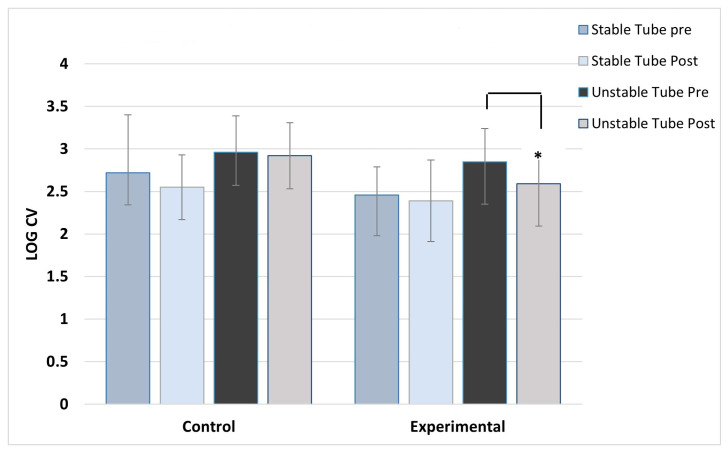
Left paraspinal activation variability. ***** indicates the experimental group’s significant post-training reduction in activation variability with unstable tube.

**Figure 6 jfmk-08-00136-f006:**
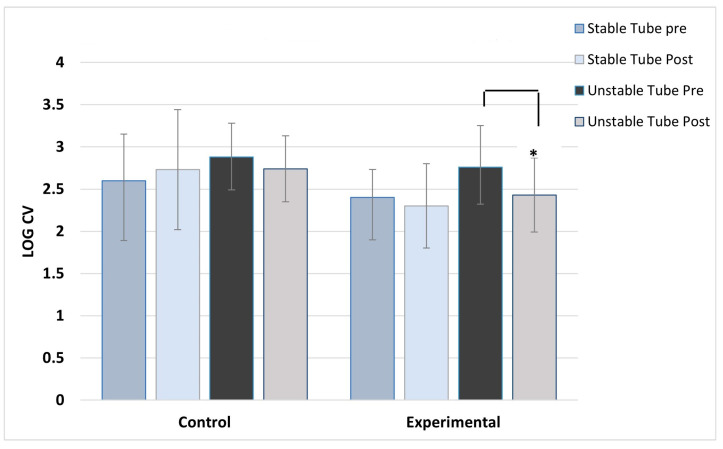
Right Vastus lateralis activation variability. ***** indicates the experimental group’s significant post-training reduction in activation variability with unstable tube.

**Figure 7 jfmk-08-00136-f007:**
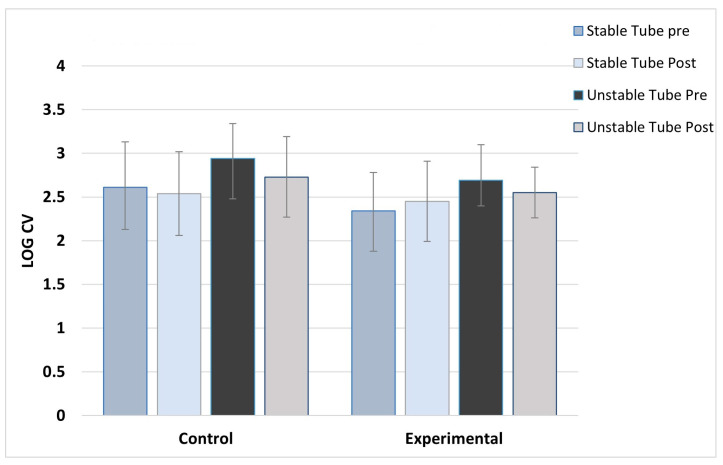
Left vastus lateralis activation variability. No significant differences.

**Table 1 jfmk-08-00136-t001:** Subject Characteristics.

	Control	Experimental
Variable	N =13 (2 Male, 11 Female)	N =17 (6 Male, 11 Female)
Age (y)	19.7 ± 0.75	19.9 ± 1.17
Height (cm)	166.5 ± 9.59	166.9 ± 11.95
Body mass (kg)	67.45 ± 13.03	68.6 ± 11.81
SBP (mmHg)	114.8 ± 11.18	115.5 ± 6.58
DBP (mmHg)	66.5 ± 6.39	71.2 ± 8.21

Data expressed as mean ± SD.

## Data Availability

Data are available in a publicly accessible repository (Open Science Framework: https://osf.io (accessed on 9 August 2023) and made available for public access at the time of manuscript publication.

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
