# Peer review of "Effect of Instability Training on Compensatory Muscle Activation during Perturbation Challenge in Young Adults"

_jfmk, 2023, doi:10.3390/jfmk8030136_

Round 1

Reviewer 1 Report

The work has scientific merit and makes it possible to advance specific knowledge for the exercise prescription with the aim of improving balance and reducing the risk of injury and falls in individuals. However, I suggest adjustments to improve the article.

L.76: In one study (2016) – Authors must include the reference number.

L. 81 and 82: A similar study - Authors should cite the study.

Were study participants trained or experienced squatters? The authors need to better present the characteristics of the participants.

Why did one group get 13 and the other 17 if the allocation was randomized? Was there a dropout? How many? If so, what were the reasons? Authors should clarify this.

L. 184 and 186: change "coefficient of variability" to “coefficient of variation”

The authors must not repeat the exposition of the results. Authors must opt for tables or graphs.

L. 234: change (11,13) to [11,13]

Authors should discuss the non-significant results in Left vastus lateralis activation variability. Why was there a reduction on the right side and not on the left side in the variability of activation of the vastus lateralis muscle?

The authors must present the limitations of the study.

Sincerely,

Author Response

Listed as Reviewer 1

L.76: In one study (2016) – Authors must include the reference number.

Edit completed. Thank you

  1. 81 and 82: A similar study - Authors should cite the study.

            Edit completed. Thank you

Were study participants trained or experienced squatters? The authors need to better present the characteristics of the participants.

The subjects were individuals who exercised 2-3 days per week, and were not weightlifters. Most participated in general fitness activities and some recreational sport activity. We did provide a familiarization period before initial testing, and the tube itself weighed only 11 kg. We have added some additional information about the subjects (lines94-97).

Why did one group get 13 and the other 17 if the allocation was randomized? Was there a dropout? How many? If so, what were the reasons? Authors should clarify this.

We have added an explanation in lines 102-107. This study took place across January 2019 and May 2023, which was the time of the COVID pandemic. As such we had some subjects lost due to stoppage of research activities.  Two control subjects withdrew and our randomization dictated that the last subjects selected we experimental, giving us a 13 control, 17 experimental subject tally for 30 total. We were not approved by our IRB to exceed 30 total subjects, thus the unbalanced sample size.

  1. 184 and 186: change "coefficient of variability" to “coefficient of variation”

Edit completed

The authors must not repeat the exposition of the results. Authors must opt for tables or graphs.

We have removed Tables 2 and 3

  1. 234: change (11,13) to [11,13]

Thank you for your sharp eyes! Edit complete

Authors should discuss the non-significant results in Left vastus lateralis activation variability. Why was there a reduction on the right side and not on the left side in the variability of activation of the vastus lateralis muscle?

This was a curious result we had discussed ourselves, but did not feel we could speculate. Please see our added information in the discussion. It is always difficult to speculate (safely), but we did add some discussion and a reference. We feel that leg dominance would be something interesting to look at for future studies in the degree of adaptation. Since the lower limbs did not receive the same degree of stimulus as the deltoids and paraspinal muscles (since the upper body absorbed much of the instability) perhaps only the dominant leg saw improvement. We did not measure limb dominance however, but one could predict the majority of subjects were right handed, so we offered some comments and a suggestion to pursue a future project with the question.

The authors must present the limitations of the study.

Thank you. We have added more information about the limitations of the study and possible avenues for future research (see highlighted discussion text)

Reviewer 2 Report

Thank you for the opportunity to review a very interesting work. Just two notes for the authors: a) blood pressure above 130/90 is mentioned as an exclusion criterion. However, the value presented in table 1 for the control group seems to indicate that some of the participants have higher systolic blood pressure values; b) what this work adds is that it is possible to obtain results in two weeks using this training technique; however, this does not eliminate the possibility that training over time and/or using other techniques would obtain similar results; In this sense, it will be important to mention the limitations of the study, as well as delve into possibilities for future work.

Author Response

Listed as Reviewer 2

Thank you for the opportunity to review a very interesting work. Just two notes for the authors:

Thank you for you review of this paper. We appreciate your insights and comments.

  1. a) blood pressure above 130/90 is mentioned as an exclusion criterion. However, the value presented in table 1 for the control group seems to indicate that some of the participants have higher systolic blood pressure values;
  2. Thank you for spotting this! We went back through all of our subject folders and found 2 errors in data entry. Resting blood pressure was sometimes taken more than once, as subjects often are agitated when “resting” in the lab settings. In cases of elevated BP beyond 130 or 90 we waited and remeasured. If it came below 130/90 we allowed the subjects to participate. We had at least 3 subjects that had multiple measures taken, but the data entry showed only the first measure. We have reviewed all measurement data of all subject characteristics and updated the Table to reflect the corrected measures. We also added a line in the methods to describe the repeat measuring when a value was above our criteria. Thank you for spotting what we clearly overlooked.

  1. b) what this work adds is that it is possible to obtain results in two weeks using this training technique; however, this does not eliminate the possibility that training over time and/or using other techniques would obtain similar results; In this sense, it will be important to mention the limitations of the study, as well as delve into possibilities for future work.

Thank you. We have edited the discussion to include more about the limitations of the study and possible avenues for future work. Please see highlighted text.

Reviewer 3 Report

Thank you for allowing me to review this study, which examined the effect of two weeks’ of front squat instability training using a water-filled training tube on force steadiness during an instability challenge.

I believe the Title should include the population in which this study was conducted and should end with the phrase “in young adults” (or similar).

The subjects of this study were young participants. Therefore, the findings of this study in terms of improved force steadiness (reductions in EMG activation variability) may not apply to clinical populations, at least within this relatively short training period of two weeks. Also, clinical populations may have a problem adhering to a specific, more physically demanding protocol and may require several modifications during its implementation. Therefore, some ‘clinical link’ will have to be made by the authors for this study, probably with similar studies of perturbation training (with water-filled training tubes or any other type of a ‘top-down’ device, if any) conducted in populations that would benefit from perturbation training.

Also, is force steadiness linked with balance improvement? And under which situations? Some information regarding the theoretical framework for conducting this study is required.

The calculation of force steadiness via the natural log of coefficient of variation has to be explained, and some relevant references are required around lines 162-3.

In the authors’ opinion, via which neuromuscular mechanism did the reductions in EMG activation variability occur? And for how long, in the authors' view, are about to last in this population? Some follow-up measurements are required to determine if the ‘effect’ is short-lived or longer-lasting, to determine the frequency in the 2-week training bouts for this and other populations.

Line 58: Do you mean ‘bottom up’?

Author Response

Listed as Reviewer 3

I believe the Title should include the population in which this study was conducted and should end with the phrase “in young adults” (or similar).

We have edited the title based on your input. Thank you

The subjects of this study were young participants. Therefore, the findings of this study in terms of improved force steadiness (reductions in EMG activation variability) may not apply to clinical populations, at least within this relatively short training period of two weeks. Also, clinical populations may have a problem adhering to a specific, more physically demanding protocol and may require several modifications during its implementation. Therefore, some ‘clinical link’ will have to be made by the authors for this study, probably with similar studies of perturbation training (with water-filled training tubes or any other type of a ‘top-down’ device, if any) conducted in populations that would benefit from perturbation training.

We have added more information in the introduction and the discussion (see highlighted text) about some clinical studies with older populations. These studies also point to the neurologic improvements seen in older populations. However much of the instability training research simply looks at the magnitude of muscle activation after training on different surfaces or using unstable loads. We feel examining force steadiness begins to look at the reaction to the instability challenges a bit more specifically (albeit still in a gross muscle activation context)

Also, is force steadiness linked with balance improvement? And under which situations? Some information regarding the theoretical framework for conducting this study is required.

We have edited the introduction (see highlighted text) to provide a more comprehensive view of the balance strategy, from proprioception to motor output and the continual adjustment needed in order to maintain stability. One reason we are interested in this water tube style of training is the continuous inputs it provides, compared to standard methods of trip inducing, or lean inducing or upper body impact perturbation.

The calculation of force steadiness via the natural log of coefficient of variation has to be explained, and some relevant references are required around lines 162-3.

We have added more information about this (now lines 172-178)

Force steadiness is essentially the variation in the integrated EMG measures across repetitions. Past research has used standard deviation as a measure [3], coefficient of variation[37] and Log of the CV [31,32]. We chose the latter, to match the methodology of our previous work.. Using the integrated EMG signal across test trial repetitions force steadiness was measured using the natural log of the percent coefficient of variation (Ln (SD/mean))

In the authors’ opinion, via which neuromuscular mechanism did the reductions in EMG activation variability occur? And for how long, in the authors' view, are about to last in this population? Some follow-up measurements are required to determine if the ‘effect’ is short-lived or longer-lasting, to determine the frequency in the 2-week training bouts for this and other populations.

 We have added some information in the discussion to offer mechanism within the context of this present study. Our measurements were not specific enough to state definitively which neuromuscular system were affected, but we offer suggestions based on the population studies and past research.

Line 58: Do you mean ‘bottom up’?

Yes! Thank you for catching that. Edit complete

Round 2

Reviewer 3 Report

Thank you for effectively addressing all my comments. The manuscript is suitable for publication.